# Variations in Diacron-Reactive Oxygen Metabolites and Biological Antioxidant Potential Across Reproductive Phases and Parities in Sows Reared Under Different Production Systems

**DOI:** 10.3390/ani15182638

**Published:** 2025-09-09

**Authors:** Shoichi Okada, Michiko Noguchi, Yosuke Sasaki, Reiichiro Sato

**Affiliations:** 1Graduate School of Medicine and Veterinary Medicine, University of Miyazaki, 1-1 Gakuen Kibanadai-nishi, Miyazaki 889-2192, Miyazaki, Japan; shoichi_okada@med.miyazaki-u.ac.jp; 2School of Veterinary Medicine, Azabu University, Fuchinobe 1-17-71, Chuo-ku, Sagamihara-shi 252-5201, Kanagawa, Japan; 3Department of Agriculture, School of Agriculture, Meiji University, Kawasaki 214-8571, Kanagawa, Japan

**Keywords:** BAP, d-ROMs, oxidative stress, reproductive score, sow

## Abstract

This study aimed to develop an available way to evaluate oxidative stress on commercial swine farms. We tested two blood measurements: d-ROMs indicate the total amount of oxidants in the body, and BAP indicates the antioxidant capacity. It was shown that a high degree of oxidative stress at farrowing negatively affected reproductive performance later on. This may be due to decreased oocyte and embryo quality under high oxidative stress. On the other hand, there were no differences in BAP values among groups, suggesting that the effects may not occur when feed and sow genetic lines are similar across farms. We also observed a trend toward reduced oxidants in the group rearing environment and higher oxidative stress in first lactation pigs. This method is easy to apply in the clinical field and may serve as a new indicator for reproductive management. Future studies should include the long-term effects of oxidative stress and the effects of antioxidant supplementation.

## 1. Introduction

Oxidative stress results from an imbalance between the antioxidant defense capacity and the production of oxidants such as reactive oxygen species (ROS). While a specific concentration of ROS is essential to maintain biological functions, excessive levels of ROS can adversely impact livestock species, including pigs, through mechanisms such as lipid peroxidation, protein modifications, and DNA oxidation [1,2], which are associated with a decline in the reproductive performance of sows [3].

Zhao Y et al. (2020) [3] stated that oxidative stress induced by a high-temperature environment was found to impair the reproductive performance in pigs, as evidenced by a reduction in the total litter size, number of live-born piglets, increased stillbirths, and decreased weight gain. Recent studies have demonstrated that oxidative stress exerts multifactorial influences on reproductive disorders in sows [4]. In late gestation, increased oxidative stress has been reported to induce placental dysfunction [4], thereby impairing the supply of nutrients and oxygen to the fetus and resulting in intrauterine growth restriction and low birth weight [5]. Moreover, oxidative stress adversely affects placental barrier function and hormonal balance, and has been associated with an increased risk of abortion and stillbirth [4,6]. Notably, in sows with larger litter sizes, oxidative stress levels tend to be further elevated [6], which may negatively impact neonatal development as well as maternal behavior during the periparturient period. In the postpartum period, oxidative stress has been implicated in the pathogenesis of postpartum dysgalactia syndrome and other reproductive disorders such as mastitis–metritis–agalactia complex and endometritis in livestock [7,8,9]. These disorders are associated with prolonged weaning-to-estrus intervals and reduced reproductive efficiency [10], underscoring the increasing importance of oxidative stress management in practical swine production systems.

It has been well-established that oxidative stress in pigs is frequently exacerbated by various factors including reduced ventilation efficiency [11], heat stress under high ambient temperatures [11,12], the ingestion of mycotoxin-contaminated feed, particularly deoxynivalenol [13,14], feed oxidation [11], and severe stress during parturition and weaning [10]. The resulting tissue damage and immunosuppression exert substantial negative effects on both reproductive performance and overall health. As a countermeasure, supplementation with antioxidant nutrients such as vitamins E and C and selenium has proven effective [15]; however, in practical settings, it is often difficult to accurately assess the degree of oxidative stress to which pigs are exposed. Consequently, uncertainty remains regarding the optimal dosage and duration of supplementation. Therefore, quantitative evaluation using oxidative stress markers is essential to establish evidence-based strategies for management and nutritional supplementation [15].

Evaluating the oxidative stress of pigs raised in farms is crucial to understand the underlying mechanisms and develop countermeasures. Despite the growing recognition of the potential impact of oxidative stress on pig health, measuring free radical levels in clinical settings remains challenging [16,17]. Free radicals have a very short half-life, typically only a few nanoseconds [18], which makes it difficult to accurately assess their concentration and activity in living organisms. Free radicals are highly reactive and can interact with various molecules, further complicating their detection and quantification [19]. Recent technological advances in the development of reliable biomarkers for oxidative stress have facilitated the clinical assessment of oxidative stress status. Diacron-reactive oxygen metabolites (d-ROMs) and biological antioxidant potential (BAP) are two biomarkers that have shown promise in human studies [20]. The d-ROM evaluation serves as a comprehensive indicator of oxidative stress by quantifying the total peroxides including hydroperoxides. These peroxides are metabolic byproducts generated when ROS and free radicals oxidize various biomolecules including lipids, proteins, amino acids, and nucleic acids. By measuring this metabolite, it is possible to assess the extent of comprehensive oxidative damage. The BAP test, on the other hand, provides a holistic assessment of antioxidant capacity by measuring the reducing power of both endogenous and exogenous antioxidants. Endogenous antioxidants include albumin, bilirubin, reduced glutathione, and uric acid, while exogenous antioxidants encompass vitamins C and E and polyphenols. This test effectively gauges the body’s overall ability to neutralize oxidative species and maintain redox homeostasis. These measurement systems do not require specialized equipment, making it highly convenient for farm use and demonstrating its practicality in field settings. Both of these tests are simple, fast, and reliable methods for evaluating oxidative stress. d-ROMs have shown good stability in blood samples stored for longer periods at −80 °C [21], suggesting their usefulness for determining the effectiveness of antioxidant therapy, assessing the risk of developing diseases associated with oxidative stress, and monitoring disease progression.

To the best of our knowledge, no previous studies have investigated these markers for the assessment of oxidative stress in pigs. The application of these biomarkers in clinical pig farms could facilitate the continuous monitoring of oxidative stress levels, potentially leading to substantial improvements in farm productivity. The deleterious impact of oxidative stress on reproductive outcomes has been well-documented in humans and various animal models. Consequently, it is hypothesized that analogous effects may be present in field conditions. Thus, this study aimed to investigate d-ROMs and BAP in relation to reproductive performance in sows.

## 2. Materials and Methods

### 2.1. Experimental Animals

The study was conducted from November 2022 to July 2023 on two commercial farms (Farm A with 1100 sows and Farm B with 1000 sows), both belonging to the same company in the Kyushu region of Japan. The study sites are located at 32°59′34″ N, 130°34′26″ E (32.9928° N, 130.5739° E; WGS84 datum) and at 33°14′06″ N, 130°36′42″ E (33.235156° N, 130.611630° E; WGS84 datum), respectively. Farm A had a group management system utilizing an open-pen approach during the gestation period, while Farm B employed an individual stall-based management system. The size of each stall was 2.1 × 0.6 m (1.26 m^2^). Twenty-eight crossbred sows (Landrace × Large White; Japan Agricultural Co., Tokyo, Japan) were observed throughout the study period including their first farrowing and the subsequent farrowing. In order to ensure the inclusion of clinically healthy sows and to eliminate potential confounding variables, the selection of experimental animals was performed by personnel at the trial farm. Inclusion criteria were that sows included in the study were clinically healthy, with no history of systemic disease within the past six months. A current veterinary examination confirmed normal vital signs and the absence of lameness or skin lesions. Exclusion criteria determined that sows were excluded if they exhibited observable lameness, displayed aggression that precluded safe handling, or were unable to stand for routine procedures. Sows were classified into two groups based on parity: a low-parity group (parity 1–2, Farm A: n = 9, Farm B: n = 9) and a high-parity group (parity 4–6, Farm A: n = 5, Farm B: n = 5). The sample size of pigs for the experiment was determined with reference to a relevant published study [6]. Blinding was not performed because of the design of this study and the unlikely effect of bias when selecting groups Any sows that developed health-related issues during the trial period were excluded from the study. During the periparturient period, all sows were provided with the same feed containing adequate minerals and vitamins (Appendix A). No antioxidant additives were included in the study. Fostering was carried out within 3 days of farrowing, and the number of suckling pigs per sow was adjusted to 9 or 10. This study was approved by the Institutional Animal Care and Use Committee, University of Miyazaki (2023-039).

### 2.2. Blood Sampling

Blood samples were collected from the jugular vein using 10-mL syringes and disposable 16-gauge × 0.1-mm hypodermic needles (NIPRO Co. Inc., Osaka, Japan) at the following three time points: farrowing (day 0), weaning (21 days after the farrowing), and pregnancy term (65 days after the farrowing). Samples were transported to the University of Miyazaki after collection under cooling conditions. The samples were centrifuged at 1500× *g* to separate serum and stored at −80 °C until analysis.

### 2.3. Measurement of Plasma d-ROMs and BAP

The frozen serum samples were thawed at 4 °C prior to measurement. The d-ROMs and BAP analyses were performed using the FREE carpe diem^®^ system (WISMERLL company Ltd., Tokyo, Japan), according to the manufacturer’s instructions. ROS/free radicals oxidize lipids, proteins, amino acids, nucleic acids, and other substances in the body and produce hydroperoxide (ROOH). To capture the total peroxides, the samples were mixed with an acetic acid buffer solution at pH 4.8. The presence of iron ions converts peroxides into radicals. The coloring solution, N,N-diethyl paraphenylenediamine, was then mixed with the radicals, and the concentration was measured. The d-ROM values were reported using Carratelli units (U.CARR.), [1 U.CARR. equivalent to 0.8 mg/L of hydrogen peroxide (H_2_O_2_)] and the BAP values were expressed in [μmol/L]. The OSI (oxidative stress index) was calculated as the ratio of oxidant to antioxidant capacity.OSI = d-ROMs/BAP

Since oxidative stress indicates an imbalance between oxidants and antioxidants, this index can be used to assess the degree of oxidative stress [22].

### 2.4. Collection of Reproduction Score

We used WebPICS (National Federation of Agricultural Cooperative Associations, Chiyoda-ku, Tokyo, Japan) to extract the following data: nursing period, total number of litters, number of live births, number of stillbirths, and weaning-to-estrus interval (WEI). Pregnant sows were subjected to pregnancy examinations using real-time ultrasound equipment fitted with a 5.0 MHz, 50-mm sector probe (Honda Electronics Co., Ltd., Toyohashi, Japan) between days 22 and 26 after artificial insemination. The presence of a fetus was assessed; if no fetus was identified, artificial insemination was promptly conducted upon the reconfirmation of estrus. In this study, all sows were confirmed to be pregnant after artificial insemination. The mean duration of lactation was 21.28 days, and the average weaning-to-estrus interval (WEI) was 8.27 days.

### 2.5. Statistical Analysis

All datasets were assessed for normality using the Kolmogorov–Simonov test and screened for potential outliers using the Smirnov–Grubb test. To investigate the relationships with reproductive performance and oxidative stress markers, we conducted multiple regression analysis. This analysis aimed to ascertain the influence of oxidative stress markers during the initial perinatal phase at parturition, weaning, and gestation on the total and normal litter sizes in the subsequent perinatal period. The dependent variables were defined as the total number of litters and the number of live births. As the number of stillbirths did not exhibit a normal distribution, it was excluded from the validation. The explanatory variables included parity (converted to a binary variable with 0 as low and 1 as high), farm (converted to a binary variable with A as 0 and B as 1), and oxidative stress markers (d-ROMs, BAP, and OSI). Due to the interrelated nature of the oxidative stress markers, each marker was analyzed separately. Thus, the explanatory variables were structured as follows for each analysis: “parity (Low or High), farm (A or B), d-ROMs”, “parity (Low or High), farm (A or B), BAP”, and “parity (Low or High), farm (A or B), OSI”. Repeated-measures analysis of variance was used to compare oxidative stress markers by parity, farm, and stage. The dependent variables were oxidative stress markers (d-ROMs, BAP, and OSI), and the independent variables were parity (Low or High), farm (A or B), and stage (farrowing, weaning, or pregnant). The repeated measure was stage, and the model was tested using the sow’s identification number as the subject term. All significant effects were tested by the Tukey–Kramer multiple-comparison test. Statistical significance was set at *p* < 0.05. Differences of less than 0.1 and ≥0.05 were considered as trends. All statistical analyses were performed using EZR (Saitama Medical Center, Jichi Medical University, Saitama, Japan), which is a graphical user interface for R (The R Foundation for Statistical Computing, Vienna, Austria) [23].

## 3. Results

### 3.1. Sow Reproductive Outcomes

The reproductive performance of all sows showed a tendency for a higher total number of piglets during the subsequent periparturient phase than the initial phase (*p* < 0.10) (Table 1). During the initial period, Farm A exhibited a significantly higher total piglet number and tended to have a higher number of live births than Farm B (*p* < 0.05) (Table 1). However, during the subsequent period, no significant differences were observed between the two characters. Moreover, there was no significant difference in reproductive performance between the parity groups (Table 1).

### 3.2. Relationship with Oxidative Stress

Multiple regression analysis revealed that at the time of parturition, both d-ROMs and OSI exhibited a significant inverse relationship with the number of total born and live born piglets (*p* < 0.05) (Table 2). Conversely, no significant associations were observed with the BAP. Furthermore, none of the oxidative stress markers measured at the initial weaning or during gestation exhibited a relationship with reproductive outcomes in the subsequent perinatal period.

### 3.3. Oxidative Stress Marker Levels and Subsequent Reproductive Score

Results of the repeated-measures analysis of variance indicated significant differences in d-ROMs and OSI across parity levels (*p* < 0.01) and in BAP across stages (*p* < 0.01) (Table 3). The d-ROMs and OSI levels were significantly higher in the low-parity group than in the high-parity group, while the BAP levels were significantly lower during the weaning stage than during the farrowing and pregnant stages (Table 3). Receiver operating characteristic (ROC) analysis was performed to predict the total number of piglets born in the subsequent farrowing.

## 4. Discussion

The present study reported the continuous monitoring of accurate and reproducible data using an automated analyzer to measure d-ROMs and BAP. To our knowledge, this study represents the first instance of measuring d-ROMs and BAP in commercial farm pigs.

This study confirmed that elevated oxidant levels at farrowing worsen subsequent reproductive outcomes (Table 2). Elevated oxidant levels during parturition were associated with litter size on subsequent farrowing. This may be caused by impaired uterine recovery and adverse effects on oocyte integrity and embryo development [4]. These findings emphasize the necessity of monitoring oxidative stress markers around farrowing as one of the predictive tools for reproductive performance.

Moreover, no significant association between oxidative stress and BAP was observed. Enhancing antioxidative capacity including vitamins C and E supplementation has been shown to reduce oxidative stress by lowering ROS levels in follicular fluid, improving oocyte quality, and increasing the fertilization and pregnancy rates [24,25]. In the present study, no differences in antioxidant capacity were observed between groups, likely due to identical feed content, limiting the ability to assess its impact on reproductive performance. Notably, the BAP levels were lower during weaning than during farrowing and pregnancy. This reduction may reflect the depletion of reducing substances used to neutralize oxidants generated at parturition. Further investigation is needed to confirm this hypothesis.

Additionally, our findings suggest that pigs raised in the group rearing system produced fewer oxidants than those raised in individual stalls. Previous studies have also analyzed the reproductive performance of sows raised in different housing systems such as individual stalls and group housing systems. Group housing allows for more natural behaviors and lower stress levels, which may contribute to improved reproductive performance [26]. In contrast, individual stalls restrict movement and natural behaviors, leading to high stress levels, although they allow for controlled feeding and reduced social stress. Studies have reported mixed results, with some indicating comparable or even better reproductive performance in sows raised in group housing systems relative to those in individual stalls [27]. Therefore, further research is warranted to determine whether the difference in the number of piglets born is actually due to the lack of behavioral restrictions associated with stalling.

Our study also showed that the low-parity group had higher values of d-ROMs than the high-parity group (Table 3). In the context of gilts, they undergo significant mammary gland development during pregnancy, which is associated with intense energy metabolism [28]. Therefore, replacement gilts sustain significantly higher oxidative conditions than multiparous sows [29]. No difference in BAP levels was observed between the two groups, suggesting that antioxidant capacity does not vary significantly with their parity or age. Previous studies have reported that antioxidant capacity varies more with dietary antioxidants and caloric intake than with aging [30]. Regardless of the differences of the degree of oxidative stress, there were no significant differences in the reproductive performance at the subsequent farrowing. Higher levels of oxygen consumption and energy metabolism observed in younger individuals compared with older individuals are physiological and are commonly seen [29], and therefore may not have adverse effects on reproductive organs due to the produced oxidants.

Our findings highlight the utility of d-ROMs and BAP as markers for monitoring oxidative stress in gilts and primiparous sows, thereby enabling the development of management protocols aimed at mitigating potential impacts on reproductive efficiency. However, several limitations should be acknowledged. This study was conducted on a limited number of farms under standardized feeding protocols, which may not fully represent diverse commercial conditions. Future research should validate these biomarkers across different genetic backgrounds and environments as well as assess cost effective, field applicable monitoring strategies along with their economic impact on swine production.

## 5. Conclusions

The results of this study indicate that d-ROMs and BAPs may serve as useful biomarkers for clinical assessing oxidative stress. These markers provide the evaluation of oxidative stress severity and its detrimental effects on sow reproductive performance.

## Figures and Tables

**Table 1 animals-15-02638-t001:** Comparison of reproductive performance among all sows, Farm A and Farm B, and low- and high-parity groups at first and subsequent farrowing.

Items	All Sows		Farm A		Farm B		Low-Parity		High-Parity	
Farrowing	First	Subsequent	First	Subsequent	First	Subsequent	First	Subsequent	First	Subsequent
Number of sows	28	28	14	14	14	14	18	10	18	10
Parity	2.32	3.32	2.28	3.56	2.35	3.56	1.28	2.28	4.2	5.2
Total born	15 (5–23)	16 (11–20) †	16.5 (8–23) a	16.5 (14–20)	14 (5–19) b	15.5 (11–20)	15 (5–21)	15.5 (11–20)	15 (7–23)	17 (13–20)
Born alive	13.5 (3–19)	14 (8–19)	14 (8–19)	14 (8–19)	13 (3–16)	14 (10–16)	14 (3–19)	14 (9–19)	13 (7–18)	14 (8–16)
Stillborn (including mummy)	1.5 (0–7)	2.5 (0–7)	1 (0–7)	3 (0–7)	2 (0–5)	1.5 (0–6)	1.5 (0–7)	1.5 (0–6)	1.5 (0–5)	3 (0–7)

Parity is shown as the means. The values for total born, born alive, and stillborn are expressed as the median (range: minimum to maximum). Dagger (†) indicates that differences in the total born number were statistically significant at *p* ≤ 0.1. Different superscript letters (a, b) indicate statistically significant differences at *p* ≤ 0.1.

**Table 2 animals-15-02638-t002:** Significances to the relationship between oxidative stress-related markers in the first periparturient period and reproductive performance in the subsequent periparturient period.

First Farrowing Explanatory Variable	Subsequent Farrowing Total Born	Regression Coefficient	Subsequent Farrowing Live Born	Regression Coefficient
Parity	*p* = 0.735		*p* = 0.518	
Farm	*p* < 0.05	−1.7774 ± 0.8393	*p* = 0.262	
d-ROMs	*p* < 0.05	−0.0058 ± 0.0021	*p <* 0.05	−0.0049 ± 0.0022
Parity	*p* = 0.311		*p* = 0.655	
Farm	*p* = 0.232		*p* = 0.777	
BAP	*p* = 0.888		*p* = 0.594	
Parity	*p* = 0.311		*p* = 0.831	
Farm	*p* < 0.05	−1.8965 ± 0.8524	*p* = 0.172	
OSI	*p* < 0.05	−21.0738 ± 7.5574	*p* < 0.05	−20.6020 ± 7.8065

Multiple regression analysis was performed to examine the relationship between oxidative stress markers (d-ROMs, BAP, OSI) at the first periparturient period and reproductive performance in the subsequent periparturient period. Explanatory variables included parity (low or high) and farm (A or B). Data are presented as regression coefficients with *p*-values. Significant associations (*p* < 0.05) are indicated.

**Table 3 animals-15-02638-t003:** Median values (min-max) of the oxidative stress markers for farm type, parity, and stage factors.

Category (Subgroup)	N	d-ROMs [U.CARR.]	BAP [μmol/L]	OSI
Farm				
A	42	1045.5 (690–1425)	3616.6 (2812.8–4404.5)	0.301 (0.192–0.445)
B	42	955.5 (627–1419)	3551.75 (2666.1–4319.8)	0.273 (0.193–0.421)
Parity				
Low	54	1116 (750–1425) a	3600.15 (2764–4319.8)	0.302 (0.206–0.445) a
High	30	930 (627–1380) b	3497.6 (2666.1–4404.5)	0.249 (0.192–0.417) b
Stage				
Farrowing	28	987 (627–1380)	3775.4 (3102.2–4404.5) a	0.253 (0.192–0.417)
Weaning	28	1024.5 (651–1425)	3367.9 (2666.1–4120.1) b	0.306 (0.193–0.445)
Pregnant	28	1015.5 (690–1419)	3580.4 (3129.1–4057.8) a	0.300 (0.206–0.425)
**Factor**		**d-ROMs [U.CARR.]**	**BAP [μmol/L]**	**OSI**
Farm		*p* = 0.122	*p* = 0.388	*p* = 0.164
Parity		*p* < 0.01	*p* = 0.336	*p* < 0.01
Stage		*p* = 0.706	*p* < 0.01	*p* = 0.085

Repeated-measures analysis of variance was used to compare the oxidative stress markers by parity, farm, and stage. The dependent variables were oxidative stress markers (d-ROMs, BAP, OSI), and the independent variables were parity (low or high), farm (A or B), and stage (farrowing, weaning, pregnancy). Data are expressed as the median (range: minimum to maximum). Different letters indicate significant differences (*p* < 0.05).

## Data Availability

The original contributions presented in the study are included in the article/Appendix A, further inquiries can be directed to the corresponding author.

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
