# Peer review of "Variations in Diacron-Reactive Oxygen Metabolites and Biological Antioxidant Potential Across Reproductive Phases and Parities in Sows Reared Under Different Production Systems"

_animals, 2025, doi:10.3390/ani15182638_

Round 1

Reviewer 1 Report

Comments and Suggestions for Authors

This manuscript presents original research exploring the potential of d-ROMs and BAP as stable biomarkers for assessing oxidative stress in breeding sows. Some findings are relevant and provide grounds for future research. The document is well-written and relatively easy to understand. However, some observations and suggestions should be addressed.

Line 23: antioxidant instead of anti-oxidant.

Materials and methods

Lines 127-128: Describe in general terms the composition of ingredients and nutrients in the diet.

Results

Title of Table 1. The sentence "Parity is shown as means. The values for total born, born alive, and stillborn are expressed as the median (range: minimum to maximum). Dagger (†) indicates that differences in the total born number was statistically significant at p ≤ 0.1.”, should be placed as a footnote to Table 1.

Title of Table 2: Significances to the relationship between......continue

Lines 214-215: Table 3 does not include P-values for OSI.

Table 3. It is recommended to delete Table 3. The P-values obtained for each marker can be inserted into three lines at the end of Table 4.

Table 4. Suggested title: Median values ​​(min-max) of oxidative stress markers for farm type, parity, and stage factors.

Lines 224-227: “Data are expressed as median (range: minimum to maximum). Different letters indicate significant differences (p <0.05).”  Place these statements as a footnote to Table 4.

Reviewer 2 Report

Comments and Suggestions for Authors

Review Comments/Report

The manuscript titled “Diacron-Reactive Oxygen Metabolites and Biological Antioxidant Potential in Sows During the Periparturient Period” aims to investigate changes in d-ROMs and BAP at various physiological stages concerning different management systems and parities in sows. The authors attempt to demonstrate how these biomarkers can serve as indicators to monitor oxidative stress at specific stages of sow production and reproduction, providing opportunities to counteract oxidative imbalances if they occur. However, several sections require clarification, methodological rigor, and contextualization to enhance the scientific merit of the findings. Below are detailed comments organized section-wise. Therefore, major revisions are required before this manuscript can be considered for publication.

Section-wise Comments:

  1. Title
    The title should be concise and informative. The term “periparturient” specifically refers to the period 2–3 weeks before parturition. However, the authors have monitored sows across various reproductive stages beyond this timeframe. Therefore, the title should be revised accordingly. A suggested revision is: “Variations in Diacron-Reactive Oxygen Metabolites and Biological Antioxidant Potential across Reproductive Phases and Parities in Sows Reared under Different Production Systems.”
  2. Simple Summary
    It should be concise and tailored for a general audience, avoiding technical jargon.
  3. Abstract
    The abstract should be more balanced. Reduce the background information and provide more specific details regarding the Materials and Methods. Ensure the conclusion is clear, focused, and scientifically robust.
  4. Introduction
    The introduction is currently verbose and lacks focus. It should be restructured to cover the following key aspects:

Reproductive complications during the periparturient, parturient, and post-parturient periods in sows.

The relationship between oxidative stress and reproductive disorders, particularly in the periparturient period.

Factors influencing oxidative stress (e.g., housing conditions, nutrition, seasonality, parity, lactation stage) and their impact on reproductive performance.

Challenges in assessing oxidative stress biomarkers in sows across production phases and advancements in current measurement techniques.

Rationale and clear objectives of the study.

  1. Materials and Methods

Specify the geographical coordinates of the study site.

Mention whether a statistical power analysis was performed to determine sample size adequacy.

Clarify the inclusion and exclusion criteria for sow selection.

Section 2.2 suggests that samples were collected at only three time points. However, the results indicate data collection across multiple events. This discrepancy should be addressed with a detailed explanation and a graphical timeline depicting sampling points.

In Section 2.4, provide comprehensive information on key production parameters, including ranges or averages for nursing period, litter size, and weaning-to-estrus interval (WEI).

  1. Results

Subheading 3.1 is redundant and should be removed.

Revise the subheadings and text under 3.2 and 3.3 for improved clarity and expression.

Review and correct the column headers in Table 2 for consistency and accuracy.

Were OSI values analyzed using repeated measures? Table 3 does not present OSI values—this needs clarification.

  1. Discussion

The discussion is excessively lengthy. Paragraphs 2 and 3 are not effectively contributing to the scientific discourse and should be revised or removed.

The manuscript lacks a section discussing limitations of the study and future perspectives. These elements should be integrated into the discussion rather than the conclusion.

  1. Conclusions

Rewrite the conclusion to summarize the main scientific findings, ensuring it reflects the core outcomes of the study.

Reviewer 3 Report

Comments and Suggestions for Authors

The submitted manuscript Diacron-reactive oxygen metabolites and biological antioxidant potential in sows during the periparturient period addresses a very interesting topic that could help prevent worsening reproduction performance in pig farming and thus avoid economic losses.

I have a few minor comments regarding the submitted manuscript that should be corrected before publication.

The Introduction is very well constructed and describes the available knowledge.
On line 54, the publication year is missing for the citation Zhao Y et al., and on line 93 the year is also missing for Pasquini et al.

The methodology section should explain in more detail why only 28 sows were included in the entire study (14 from farm A and 14 from farm B). Both farms have a much larger sow capacity. I am not claiming that the number of animals is insufficient, but this needs to be clearly explained. At the same time, the selection mechanism of the experimental sows should be well justified.

The results are described clearly and comprehensively.
In the discussion, it should at least be indicated how the findings can be applied to improve the reproductive performance of sows.

Round 2

Reviewer 1 Report

Comments and Suggestions for Authors

The observations and recommendations were addressed in this revised version.
I only have one recommendation: Insert three-digit P values ​​in Tables 2 and 3.

Author Response

Comment 1 : The observations and recommendations were addressed in this revised version.
I only have one recommendation: Insert three-digit P values ​​in Tables 2 and 3.

Response 1: We completely agree with this comment,. And, we have revised the digit of P values in Tables. (Line 231-232, 248-249)

Reviewer 2 Report

Comments and Suggestions for Authors

The revised manuscript shows substantial improvement in clarity, structure, and scientific rigor. The concerns raised in original submission, have been adequately addressed through thoughtful revisions, improved methodological transparency, better data presentation, and more focused discussion. The revised version appears suitable for publication.

Author Response

Comment 1: The revised manuscript shows substantial improvement in clarity, structure, and scientific rigor. The concerns raised in original submission, have been adequately addressed through thoughtful revisions, improved methodological transparency, better data presentation, and more focused discussion. The revised version appears suitable for publication.

Response 1: We sincerely appreciate the reviewer’s positive evaluation and constructive feedback

Reviewer 3 Report

Comments and Suggestions for Authors

The revised manuscript has been significantly improved and is now suitable for publication in the Animals.

Author Response

Comment 1:The revised manuscript has been significantly improved and is now suitable for publication in the Animals.

Response 1: We sincerely appreciate the reviewer’s positive evaluation and constructive feedback